

# Mapping a dissipative quantum spin chain onto a generalized Coulomb gas

Oscar Bouverot-Dupuis

Université Paris-Saclay, CNRS, LPTMS, 91405, Orsay, France
Université Paris-Saclay, CNRS, CEA, IPhT, 91191, Gif-sur-Yvette, France

oscar.bouverot@ens.psl.eu

## Abstract

An XXZ spin chain at zero magnetization is subject to spatially correlated baths acting as dissipation. We show that the low-energy excitations of this model are described by a dissipative sine-Gordon field theory, i.e. a sine-Gordon action with an additional long-range interaction emerging from dissipation. The field theory is then exactly mapped onto a generalized Coulomb gas which, in addition to the usual integer charges, displays half-integer charges that originate from the dissipative baths. These new charges come in pairs linked by a charge independent logarithmic interaction. In the Coulomb gas picture, we identify a Berezinsky–Kosterlitz–Thouless-like phase transition corresponding to the binding of charges and derive the associated perturbative renormalization group equations. For superohmic baths, the transition is due to the binding of the integer charges, while for subohmic baths, it is due to the binding of the half-integer charges, thereby signaling a dissipation-induced transition.

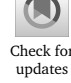
# 1 Introduction

The Berezinsky–Kosterlitz–Thouless (BKT) transition [1–4] is perhaps the most famous example of a transition which cannot be described by the Ginzburg–Landau paradigm. This transition manifests itself in the topological properties of two-dimensional systems with a two-component $O(2)$-symmetric order parameter. In particular, it admits no spontaneous breaking of the continuous symmetry as demanded by the Mermin–Wagner theorem [5]. The BKT transition is often associated with different 2D models, including the XY model, the sine-Gordon field theory, and the Coulomb gas which, each, offer different insights on the transition. In the XY model, a low-$T$ critical phase dominated by spin wave configurations is destroyed by the proliferation of topological defects, namely vortices, at sufficiently high temperature. In the Coulomb gas picture, these vortices correspond to charged particles which undergo a binding transition and are free in the high temperature phase. Using the mappings between these two models and the sine-Gordon field theory, many generalizations of the BKT transition have been proposed and gathered under the name of BKT-*like* phase transitions. To cite just a few examples of systems belonging to this universality class, XY models with nematic order corresponding to gases with fractional charges [6–9], XY models with long-range [10, 11] or anisotropic interactions [12], generalized sine-Gordon models [13], have been proposed.

Exiting the realm of classical statistical mechanics, one-dimensional quantum systems have appeared as a class of systems which can exhibit such BKT-like transitions. According to the canonical Euclidean path integral representation of 1D quantum systems [14, 15], they correspond to 2D classical systems, thus making them ideal candidates to observe the BKT phenomenology. In the field of open quantum systems, one-dimensional interacting fermionic systems with quenched disorder have, for example, been investigated using a mapping to a Coulomb gas as for the usual BKT transition [16–18]. Harnessing the techniques used in statistical physics to describe the BKT transition, Giamarchi and Schulz showed already 30 years ago that such systems exhibit a localization transition at zero temperature [16, 19], thereby extending the pioneering work of Anderson on single-particle localization [20].

The focus of this paper is on *dissipative* 1D quantum systems. It turns our that introducing dissipation amounts to replacing the previously mentioned quenched disorder with dynamical (annealed) disorder. Following the formalism developed by Caldeira, Leggett et al. [21–25], dissipation is introduced by coupling to a bath of phonons that is simple enough to be traced out, thus yielding an effective description of the system of interest. Although this type of bath

was first studied by putting it in contact with a single degree of freedom such as a spin in the spin-boson model [26,27], several many-body 1D systems were later proposed, ranging from spin chains [28–33] and 1D bosons [34], to glassy systems [35,36], and with various system-bath couplings [37,38]. In this work, we concentrate on the well-known XXZ spin chain at zero magnetization and temperature. Each spin of the chain is linked to a phonon bath *à la* Caldeira and Leggett, which can be seen as a many-body extension of the spin-boson model. The baths are furthermore spatially correlated by bath-bath nearest neighbor interactions. To the best of the author's knowledge, it is the first time that such correlated dissipation is studied, previous studies having concentrated on local independent baths [31–33] or a single global bath [39]. For a spin chain at zero magnetization and with uncorrelated local baths, it has been proposed in [33] that the system presents a BKT-like transition which is driven by dissipation for subohmic baths. In this article, we broaden the scope of this study by considering correlated baths. In order to fully understand the nature of the transition in our system, we follow an approach based on a mapping to a generalized Coulomb gas. Within this gas picture, we show that for superohmic baths, the transition is the exact BKT transition while, for subohmic baths, dissipation induces a BKT-like transition. It is worth mentioning that dissipative systems such as a single [40] or an array [41,42] of resistively shunted Josephson junctions have already been studied using approximate mappings to 1D or 2D gases with logarithmic interactions. The gaseous picture also arises naturally in the context of quantum Monte-Carlo studies of dissipative spin chains [27] where the particles are the kinks and anti-kinks of world line configurations of spins. However, our work differs from these in two ways. First, our mapping does not require any approximation and is thus mathematically exact and, second, our baths are spatially correlated and thus provide a larger variety of dissipation-induced particle-particle interactions in the gas picture.

The manuscript is organized as follows. Section 2 gives a concise summary of the main results which are derived in the following sections. Section 3 presents the microscopic model and shows how its low-energy behavior can be described by a dissipative sine-Gordon field theory. In section 4, this field theory is mapped onto a generalized 2D Coulomb gas and its basic features are described. The core of the article, identifying the BKT-like transition, is done in section 5. Finally, a brief discussion of the results and concluding remarks are made in section 6. Some technical details have been relegated in appendices A.1 to E. Throughout this article, we set $\hbar = k_B = 1$.

## 2 Main results

An XXZ spin chain at zero magnetization and zero temperature is linked to spatially correlated baths acting as dissipation (see Fig. 1). The nature of the baths is characterized by an exponent $s > 0$. Through bosonization, the low-energy physics of this system is described by a sine-Gordon action with a bath-induced long-range (in space $x$ and imaginary-time $\tau$) interaction (see Eq. (14)). The spin chain itself is essentially described by a Luttinger parameter $K$ and a velocity $u$ while the correlation between baths is encoded in a characteristic speed $v$. Taking this field theory as a starting point for our analysis, it is studied by mapping it onto a generalized 2D Coulomb gas that consists of the following particles,

- $g^{\pm}$ particles with charge $\pm 1$ which only interact through the Coulomb potential,

- $\alpha^+, \alpha^0, \alpha^-$ pairs of particles with, respectively, charges $(+1/2, +1/2)$, $(+1/2, -1/2)$, $(-1/2, -1/2)$. All these particles interact through the Coulomb potential and two particles from a same pair feel an extra charge-independent attraction. These half-integer charges are reminiscent of nematic order in generalized XY models [6–8].

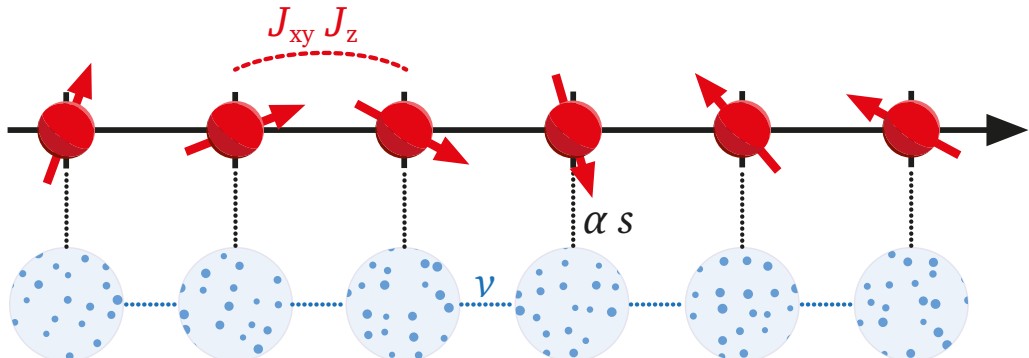

Figure 1: Schematic representation of the model. An XXZ spin chain with parameters $J_z, J_{xy}$ has its spins coupled to identical collections of harmonic oscillators acting as baths. These baths interact with their nearest neighbors with a coupling $\nu$. The spin-bath interaction is assumed to be fully described by the parameters $\alpha$ and $s$.

Two charges $q, q'$ at positions $r = (x, u\tau)$ and $r' = (x', u\tau')$ interact through

- the Coulomb potential $V_c(r, r') \sim -8Kqq' \log|r - r'|$,

- a bath-induced interaction $V_\alpha(r, r') \sim (2+s) \log|(\tau - \tau')^2 + (x - x')^2/\nu^2|$ if both particles belong to the same $\alpha$ pair.

In this Coulomb gas picture, we infer the zero-temperature ($\beta = 1/T \to \infty$) and thermodynamic ($L \to \infty$) phase diagram of the spin chain. A BKT-like phase transition is identified as the binding of particles, and the associated RG equations are derived.

- For superohmic baths ($s > 1$), the transition is at $K_c = 1/2$ and corresponds to the binding of $g^\pm$ particles. This is the usual BKT transition of the sine-Gordon model, so dissipation does not play a major role here.

- For subohmic baths ($s < 1$), dissipation shifts the transition to $K_c = 1 - s/2$ and corresponds to the binding of $\alpha$ particle pairs. This is a BKT-like transition with anisotropic RG equations in the 2D gas space when $u/\nu \neq 1$.

This transition separates two phases: a Luttinger liquid at $K > K_c$ described by a Gaussian field theory corresponding to all particles binding; and a dissipative phase at $K < K_c$, where some particles are free and screen the Coulomb interaction which acquires an exponential fallout at large distances. This decay is characterized by a dissipation length $\xi$ which diverges at the transition.

## 3 Model

In this section, we construct a field theory describing the low-energy physics of an XXZ spin chain subject to correlated dissipation. This field theory will turn out to be a dissipative 2D sine-Gordon model and then serve as the starting point for the mapping to a generalized Coulomb gas in section 4. The model we start from is a periodic XXZ spin chain of $N$ spins, total length $L$ and lattice spacing $a = N/L$, which is described by the Hamiltonian

$$H_S = \sum_{j=1}^{N} J_z S_j^z S_{j+1}^z - J_{xy} \left( S_j^x S_{j+1}^x + S_j^y S_{j+1}^y \right), \tag{1}$$

where $S_i^\mu$ are the spin-1/2 operators. Dissipation is introduced by coupling each spin to a set of harmonic oscillators through

$$H_{SB} = \sum_{j=1}^{N} S_j^z (-1)^j \sum_\gamma \lambda_\gamma X_{\gamma,j}, \tag{2}$$

where $\lambda_\gamma$ is the spin-bath coupling, and the harmonic oscillators internal dynamics is described by

$$H_{B} = \sum_{j=1}^{N} \sum_\gamma \frac{P_{\gamma,j}^2}{2m_\gamma} + \frac{1}{2} m_\gamma v^2 \left( \frac{X_{\gamma,j} - X_{\gamma,j+1}}{a} \right)^2 + \frac{1}{2} m_\gamma \Omega_\gamma^2 X_{\gamma,j}^2, \tag{3}$$

where $m_\gamma, \Omega_\gamma$ are the mass and frequency of the oscillators, and $v$ quantifies the nearest neighbor interaction between baths. Hence the total Hamiltonian of the dissipative system $H = H_S + H_{SB} + H_B$. This way of implementing dissipation is a generalization of [31–33] which considered $H_B$ with $v = 0$, i.e. local dissipation. The $(-1)^j$ coupling between the bath and the spin chain has been chosen as to induce an antiferromagnetic effect on the spin chain. Notice that this is not needed if $v = 0$ as one can then simply redefine $\tilde{X}_{\gamma,j} = (-1)^j X_{\gamma,j}$ and $\tilde{P}_{\gamma,j} = (-1)^j P_{\gamma,j}$ which absorbs the $(-1)^j$ in $H_{SB}$ and leads directly to the model studied in [33]. Other spin-bath couplings are also possible and have been studied in, for example, [37,38].

Following the pioneering work of Caldeira and Leggett [26], the effect of the bath can be fully characterized by the spectral function

$$J(\Omega) = \frac{\pi}{2} \sum_\gamma \frac{\lambda_\gamma^2}{\Omega_\gamma m_\gamma} \delta(\Omega - \Omega_\gamma). \tag{4}$$

Assuming a large number of oscillators, it is expected that the spectral function can be accurately replaced by the smooth ansatz

$$J(\Omega) \sim \alpha \tau_c^{s-1} \Omega^s, \quad \text{for} \quad \Omega \in [0, 1/\tau_c], \tag{5}$$

where $s > 0$ is the bath exponent, $\alpha$ measures the strength of the coupling to the bath, and $1/\tau_c$ is a frequency cutoff above which $J(\Omega) = 0$. The exact numerical prefactor in Eq. (5) is unimportant and can be found in appendix A.1. This form of $J(\Omega)$ accounts for many types of environments, ranging from acoustic-phonon baths to Kondo physics [26]. Following the classification from Caldeira and Leggett, we call $s = 1$ an ohmic bath, while $0 < s < 1$ is a subohmic bath and $s > 1$ is a superohmic bath. The rest of this section will be dedicated to deriving an effective action for the system. The procedure is similar to that outlined in [31–33] and relies on two key steps: first bosonizing the spin chain, and then integrating out the bath degrees of freedom.

## 3.1 Bosonization

Bosonization, as described in [43–45], allows to fully describes a fermionic system in terms of collective bosonic excitations. In order to apply it to our model, we first map our spin chain onto a spinless fermionic system using the Jordan–Wigner transformation. With the usual notations $c_j, c_j^\dagger$ for the ladder operators and $n_j = c_j^\dagger c_j$, the XXZ spin chain Hamiltonian becomes $H_S = H_{xy} + H_z$, where

$$H_{xy} = -\frac{J_{xy}}{2} \sum_{j=1}^{N} \left( c_{j+1}^\dagger c_j + c_j^\dagger c_{j+1} \right), \tag{6}$$

$$H_z = \sum_{j=1}^{N} J_z \left( n_j - \frac{1}{2} \right) \left( n_{j+1} - \frac{1}{2} \right). \tag{7}$$

In the following, we focus on the zero-magnetization sector of the XXZ spin chain, which, in our new fermionic language, corresponds to a half-filled system and a Fermi momentum $k_F = \frac{\pi}{2a}$.[1] The bosonization procedure then requires linearizing the spectrum $\varepsilon_k = -J_{xy}\cos(ka)$ of $H_{xy}$ about $k_F$. Using the usual bosonization formulas along with the canonical equilibrium path integral formalism [14,15] shows that the XXZ spin chain can be mapped onto a field theory with action $S = S_{LL} + S_g$, such that

$$S_{LL} = \int dx d\tau \frac{1}{2\pi K}\left[u(\partial_x \phi(x,\tau))^2 + \frac{1}{u}(\partial_\tau \phi(x,\tau))^2\right], \qquad (8)$$

$$S_g = -\frac{gu}{2\pi^2 a^2}\int dx d\tau \cos[4\phi(x,\tau)], \qquad (9)$$

where $x \in [0, L]$ denotes the spatial coordinate, $\tau \in [0, \beta]$ is the imaginary time coordinate, and $\phi(x,\tau)$ is a bosonic field. From now on, the zero temperature ($\beta \to \infty$) and thermodynamic ($L \to \infty$) limits will always be understood. In Eq. (8), $S_{LL}$ is the so-called Luttinger Liquid action which is ubiquitous in the low-energy description of one-dimensional quantum systems [43], and $S_g$ describes the internal interactions of the spin chain at zero magnetization. The parameter $u$ is a velocity, $K$ is the so-called Luttinger parameter and, with $g$, they are related to the microscopic parameters. The Bethe ansatz [46] gives their exact expressions, e.g. $K^{-1}_{Bethe} = \frac{2}{\pi}\arccos(-J_z/J_{xy})$, while bosonization yields approximate expressions valid in the $J_z \ll J_{xy}$ limit, e.g. $K^{-1}_{bosonization} = \sqrt{1 + 4J_z/(\pi J_{xy})}$. In addition to this description of the isolated spin chain, one needs to consider the effect of the bath captured by $H_{SB} + H_B$. Resorting again to the path integral formalism, one can show that the entire system is described by the action $S = S_{LL} + S_g + S_{SB} + S_B$, where

$$S_B = \sum_\gamma \int dx d\tau \frac{m_\gamma}{2a}X_\gamma(x,\tau)(-\partial_\tau^2 - v^2\partial_x^2 + \Omega_\gamma^2)X_\gamma(x,\tau), \qquad (10)$$

$$S_{SB} = \sum_\gamma \int dx d\tau \frac{\lambda_\nu}{a}(-1)^{x/a}X_\gamma(x,\tau)\left(-\frac{a}{\pi}\partial_x \phi(x,\tau) + \frac{(-1)^{x/a}}{\pi}\cos[2\phi(x,\tau)]\right), \qquad (11)$$

are the only terms including contributions from the bath degrees of freedom.

## 3.2 Integrating out the bath degrees of freedom

The action in Eqs. (10,11) being quadratic in the bath degrees of freedom $\{X_\gamma(x,\tau)\}$, it is possible to exactly integrate them out. This creates an effective long-range interaction mediated by the bath kernel

$$\mathcal{K}(x,\tau) = \frac{1}{\pi^2 v}\int_0^\infty d\Omega J(\Omega)\Omega K_0(\Omega\sqrt{\tau^2 + x^2/v^2}), \qquad (12)$$

where $K_0$ is a modified Bessel function of the second kind, and we have used the definition of the spectral function given in Eq. (4). Upon using the low-energy behavior of $J(\Omega)$ given in Eq. (5), one can show that $\mathcal{K}(x,\tau) \simeq \alpha\tau_c^{s-1}\left(\tau^2 + x^2/v^2\right)^{-1-s/2}/u$ for $\sqrt{\tau^2 + x^2/v^2} \gg \tau_c$ (see appendix A.1). Putting everything together, one arrives at the following effective interaction

$$S_\alpha = -\frac{\alpha\tau_c^{s-1}}{2\pi^2 au}\int_{\sqrt{(\tau-\tau')^2+(x-x')^2/v^2}>\tau_c} dx dx' d\tau d\tau' \quad \left(a(-1)^{x/a}\partial_x \phi(x,\tau) - \cos[2\phi(x,\tau)]\right)$$

$$\times\left[(\tau-\tau')^2 + \frac{(x-x')^2}{v^2}\right]^{-1-s/2}\left(a(-1)^{x'/a}\partial_{x'}\phi(x',\tau') - \cos[2\phi(x',\tau')]\right). \qquad (13)$$

---

[1] This means that the Fermi wavelength $\lambda_F = 4a$ is commensurate with the lattice spacing $a$.

For the sake of simplicity, we relate the imaginary-time and space cutoffs as $a = u\tau_c$ which does not change the underlying low-energy physics. Eq. (13) can be further simplified by dropping all the terms rapidly fluctuating as $(-1)^{x/a}$ and $(-1)^{x'/a}$. The full bosonized action $S = S_{\mathrm{LL}} + S_g + S_\alpha$ is finally given by

$$S = \int \mathrm{d}x\mathrm{d}\tau \frac{1}{2\pi K}\left[u(\partial_x\phi(x,\tau))^2 + \frac{1}{u}(\partial_\tau\phi(x,\tau))^2\right] - \frac{g}{2\pi^2 a\tau_c}\int \mathrm{d}x\mathrm{d}\tau \cos[4\phi(x,\tau)]$$

$$- \frac{\alpha}{2\pi^2 a^2\tau_c^2}\int_{\sqrt{(\tau-\tau')^2+(x-x')^2/v^2}>\tau_c} \mathrm{d}x\mathrm{d}x'\mathrm{d}\tau\mathrm{d}\tau' \cos[2\phi(x,\tau)]\cos[2\phi(x',\tau')]\mathcal{D}(x-x',\tau-\tau'), \quad (14)$$

where $\mathcal{D}(x,\tau) = \left[(\tau/\tau_c)^2 + (u/v)^2(x/a)^2\right]^{-1-s/2}$ is the dimensionless dissipative kernel. This effective action is that of a Luttinger liquid (LL) with two types of interactions: a local interaction describing the spin chain's internal dynamics and controlled by $g$, and a long-range bath-induced interaction controlled by $\alpha$. This dissipative interaction can be thought of as the bath storing information about a spin at position $x$ and time $\tau$, propagating it over a distance $x - x'$ during a period $\tau - \tau'$, and giving it back at time $\tau'$ to the spin at position $x'$. Notice that this very-strong non-Markovian effect was obtained through the exact integration of the bath and is a feature which cannot be reproduced by usual open quantum system techniques such as the Lindblad master equation. As mentioned previously, the effective action written in Eq. (14) has already been studied in the $v = 0$ case [33] which corresponds to the limit of uncorrelated baths. In this case, the last term in Eq. (14) gets modified by changing the kernel into $\mathcal{D}(x,\tau) \propto \delta(x)/|\tau|^{1+s}$ (see Appendix. A.2 for the derivation). On the other hand, the case of $v \gg 1$ is new and corresponds to the limit of Markovian, but spatially correlated, baths. In this limit, the bath-induced kernel becomes $\mathcal{D}(x,\tau) \propto \delta(\tau)/|x|^{1+s}$. These two limits for $v$ formally yield identical models up to an exchange of space and time (and a redefinition of $\alpha$ to absorb extra factors of $v/u$). Another remarkable model is achieved when $v = u$ which corresponds to an isotropic system enjoying an $O(2)$ symmetry in $(x, u\tau)$ space. For the majority of this paper we will mainly deal with finite $v$ and postpone the discussion of the limiting cases ($v$ small or large) to section 6.

## 4 Mapping to a generalized Coulomb gas

This section presents what is at the core of this article: mapping the effective action of Eq. (14) onto a 2D gas of charged classical particles. Before giving an explicit and rigorous derivation of the mapping, let us present in simple terms what is expected. Setting $\alpha = 0$ in Eq. (14) recovers the sine-Gordon action which is known to correspond to a Coulomb gas of particles with charge $\pm 1$, fugacity $z_g = \frac{g}{4\pi^2}$ and size (area) $\sigma = a^2$. We shall call these particles $g^+$ and $g^-$ depending on the sign of their charge. This result can actually be directly read out from the interaction by writing

$$\frac{g}{2\pi^2 a^2}\int \mathrm{d}^2 r \cos(4\phi(r)) = \int \frac{\mathrm{d}^2 r}{\sigma}z_g\left(e^{i4\phi(r)} + e^{-i4\phi(r)}\right), \quad (15)$$

where $r = (x, u\tau)$. This makes apparent that a particle with charge $q$ corresponds to the operator $e^{i4q\phi}$ in the field theory. Following this idea, the bath-induced term can be written as

$$\frac{\alpha}{2\pi^2 a^4}\int \mathrm{d}^2 r\mathrm{d}^2 r' \cos[2\phi(r)]\cos[2\phi(r')]\mathcal{D}(r-r')$$

$$= \int \frac{\mathrm{d}^2 r\mathrm{d}^2 r'}{\sigma^2}\left[z_{\alpha^+}e^{i2\phi(r)+i2\phi(r')} + z_{\alpha^0}e^{i2\phi(r)-i2\phi(r')} + z_{\alpha^-}e^{-i2\phi(r)-i2\phi(r')}\right]\mathcal{D}(r-r'), \quad (16)$$

where we have defined $z_{\alpha^+} = z_{\alpha^-} = \frac{\alpha}{8\pi^2}$, $z_{\alpha^0} = \frac{\alpha}{4\pi^2}$ and written $\mathcal{D}(r = (x, \tau)) = \mathcal{D}(x, \tau)$ for the sake of readability. This shows that the bath creates three distinct pairs of particles that we shall call $\alpha^+$, $\alpha^0$ and $\alpha^-$ pairs. These pairs respectively have charges $(+1/2, +1/2)$, $(+1/2, -1/2)$, $(-1/2, -1/2)$, and fugacities $z_{\alpha^+}$, $z_{\alpha^0}$, $z_{\alpha^-}$. Each particle within a pair has size $\sigma$. Moreover, it can be guessed that two particles within the same pair will experience an additional force coming from the kernel $\mathcal{D}(r - r')$.

In the following subsections, we first rigorously establish the mapping to a generalized Coulomb gas, and then describe the gas in more physical terms. The first subsection is quite technical and can be skipped at first reading as all necessary information is reminded in the second subsection.

## 4.1 Proof of the mapping

We now move on to performing the actual mapping. As for the regular sine-Gordon to Coulomb gas mapping, one starts by expanding the partition function of the field theory as

$$
\begin{aligned}
Z &= \int \mathcal{D}[\phi] e^{-S_{\text{LL}}[\phi] - S_g[\phi] - S_\alpha[\phi]} \\
&= Z_{\text{LL}} \left\langle e^{-S_g[\phi] - S_\alpha[\phi]} \right\rangle_{\text{LL}} \\
&= Z_{\text{LL}} \sum_{n=0}^{\infty} \frac{1}{n!} \left\langle \left( -S_g[\phi] - S_\alpha[\phi] \right)^n \right\rangle_{\text{LL}},
\end{aligned}
\tag{17}
$$

where $Z_{\text{LL}}$ is the partition function associated to the Luttinger liquid action $S_{\text{LL}}$. According to the shift symmetry of $S_{\text{LL}}$, only neutral particle configurations (in the gas picture) survive the average $\langle . \rangle_{\text{LL}}$ (see appendix B for more details). We therefore have to extract all neutral configurations from $\left( -S_g[\phi] - S_\alpha[\phi] \right)^n$. Let us call $n_{g^+}, n_{g^-}$ the number of $g^+, g^-$ particles, and $n_{\alpha^+}, n_{\alpha^0}, n_{\alpha^-}$ the number of $\alpha^+, \alpha^0, \alpha^-$ pairs. These are related by the fixed sum $n = n_{g^+} + n_{g^-} + n_{\alpha^+} + n_{\alpha^0} + n_{\alpha^-}$ and by the charge neutrality condition $n_{g^+} + n_{\alpha^+} - n_{\alpha^-} - n_{g^-} = 0$. Taking care of the combinatorial factor, the partition function therefore reads

$$
\frac{Z}{Z_{\text{LL}}} = \sum_{n=0}^{\infty} \sum_{\{n_i\}}^{(n)} \frac{z_g^{n_{g^+} + n_{g^-}} z_{\alpha^+}^{n_{\alpha^+}} z_{\alpha^0}^{n_{\alpha^0}} z_{\alpha^-}^{n_{\alpha^-}}}{n_{g^+}! n_{g^-}! n_{\alpha^+}! n_{\alpha^0}! n_{\alpha^-}!}
\tag{18}
$$

$$
\times \left\langle \left( \int \frac{d^2 r}{\sigma} e^{i4\phi(r)} \right)^{n_{g^+}} \left( \int \frac{d^2 r}{\sigma} e^{-i4\phi(r)} \right)^{n_{g^-}} \left( \int \frac{d^2 r d^2 r'}{\sigma^2} \mathcal{D}(r - r') e^{i2(\phi(r) + \phi(r'))} \right)^{n_{\alpha^+}} \right.
$$

$$
\left. \times \left( \int \frac{d^2 r d^2 r'}{\sigma^2} \mathcal{D}(r - r') e^{i2(\phi(r) - \phi(r'))} \right)^{n_{\alpha^0}} \left( \int \frac{d^2 r d^2 r'}{\sigma^2} \mathcal{D}(r - r') e^{-i2(\phi(r) + \phi(r'))} \right)^{n_{\alpha^-}} \right\rangle_{\text{LL}},
$$

where $\sum_{\{n_i\}}^{(n)}$ indicates a sum over all particle numbers with vanishing charge and adding up to $n$. To simplify this expression, we label by $j = 1, 2, \ldots$ all the particles of the gas and denote their position and charge by $(r_j, q_j)$. We also introduce the set of all particle pairs belonging to the same $\alpha$ pair, i.e. $E_\alpha = \{(i, j) \mid \text{particles } i \text{ and } j \text{ are from the same } \alpha \text{ object}\}$. This leads to

$$
\frac{Z}{Z_{\text{LL}}} = \sum_{n=0}^{\infty} \sum_{\{n_i\}}^{(n)} \frac{z_g^{n_{g^+} + n_{g^-}} z_{\alpha^+}^{n_{\alpha^+}} z_{\alpha^0}^{n_{\alpha^0}} z_{\alpha^-}^{n_{\alpha^-}}}{n_{g^+}! n_{g^-}! n_{\alpha^+}! n_{\alpha^0}! n_{\alpha^-}!} \prod_j \int \frac{d^2 r_j}{\sigma} \prod_{(i,j) \in E_\alpha} \mathcal{D}(r_i - r_j) \left\langle \prod_j e^{i4q_j \phi(r_j)} \right\rangle_{\text{LL}}.
\tag{19}
$$

The average is easily computed using properties of the Gaussian action $S_{\text{LL}}$ (see appendix B)

$$
\left\langle \prod_j e^{i4q_j \phi(r_j)} \right\rangle_{\text{LL}} = \exp\left[ -\beta_{\text{gas}} \frac{1}{2} \int_{|r-r'| > a} d^2 r d^2 r' n_c(r) V_c(r - r') n_c(r') \right],
\tag{20}
$$

where we have introduced the charge density $n_c(r) = \sum_j q_j \delta(r - r_j)$, the Coulomb interaction $V_c(r) = -8KT_{\text{gas}} \log \left| \frac{r}{a} \right|$ and the gas inverse temperature $\beta_{\text{gas}} = 1/T_{\text{gas}}$ (which is not to be confused with the one of the original quantum model).[2] The bath kernel $\mathcal{D}(r)$ in Eq. (19) can also be re-exponentiated to give rise to an interaction $V_\alpha(r) = -T_{\text{gas}} \log \mathcal{D}(r)$ that couples to the two-point density of $\alpha$ pairs $n_\alpha(r, r') = \sum_{(r_i, r_j) \in E_\alpha} \left[ \delta(r - r_i)\delta(r' - r_j) + \delta(r - r_j)\delta(r' - r_i) \right]$. Putting everything together gives the generalized Coulomb gas partition function

$$
Z_{\text{gas}} = \frac{Z}{Z_{\text{LL}}} = \sum_{n=0}^\infty \sum_{\{n_i\}}^{(n)} \frac{z_g^{\,n_{g^+} + n_{g^-}} z_{\alpha^+}^{\,n_{\alpha^+}} z_{\alpha^0}^{\,n_{\alpha^0}} z_{\alpha^-}^{\,n_{\alpha^-}}}{n_{g^+}! n_{g^-}! n_{\alpha^+}! n_{\alpha^0}! n_{\alpha^-}!} \prod_j \int \frac{d^2 r_j}{\sigma}
$$

$$
\times \exp\left[ -\beta_{\text{gas}} \frac{1}{2} \int\limits_{|r - r'| > a} d^2 r\, d^2 r' \left( n_c(r) V_c(r - r') n_c(r') + n_\alpha(r, r') V_\alpha(r - r') \right) \right]. \quad (21)
$$

The next subsection is devoted to the description and discussion of the gas properties.

## 4.2 Description of the Coulomb gas

The partition function (21) is the grand canonical partition function of a neutral 2D gas containing different species of particles. Figure 2 depicts a snapshot of the gas. First are the $g^+, g^-$ particles which have charge $\pm 1$ and fugacity $z_g \propto g$. These are the usual particles of the Coulomb gas. Next are exotic objects that stem from the interaction with the bath. These are pairs of particles of charge $\pm 1/2$. The three possible pairings are called $\alpha^+, \alpha^0, \alpha^-$ and, respectively, have charge $(+1/2, +1/2), (+1/2, -1/2), (-1/2, -1/2)$ and fugacity $z_{\alpha^+}, z_{\alpha^0}, z_{\alpha^-} \propto \alpha$. These particles are all subject, even within an $\alpha$ pair, to the 2D Coulomb interaction defined as

$$
\beta_{\text{gas}} q q' V_c(r, r') = -8K q q' \log \left| \frac{r - r'}{a} \right|, \quad (22)
$$

where $\beta_{\text{gas}} = 1/T_{\text{gas}}$ is the inverse temperature of the gas, and $q, q'$ are the particle charges. From Eq. (22) it appears that the vacuum permittivity $\varepsilon_0 \propto K^{-1}$. On top of that, two particles within the same $\alpha$ pair experience an extra interaction given by

$$
\beta_{\text{gas}} V_\alpha(r, r') = (2 + s) \log \sqrt{\left( \frac{\tau - \tau'}{\tau_c} \right)^2 + \left( \frac{u}{v} \right)^2 \left( \frac{x - x'}{a} \right)^2}. \quad (23)
$$

This interaction is always charge-independent and attractive.

The most striking feature of this gas is perhaps the presence of fractional charges. Indeed, if one thinks of the usual Coulomb gas particles in terms of vortices in the XY model, a particle with integer charge $q$ is associated with a vortex of winding number $q$. Going once around this vortex, the $O(2)$ spins of the XY model pick up an extra $2\pi q$ phase. If one instead considers a half-integer charge $q/2$ in the generalized Coulomb gas, this should correspond to an extra $\pi q$ phase which is reminiscent of nematic ordering, i.e. spins aligning irrespective of direction. This type of crossover between spin and nematic ordering has been studied in the context of generalized XY models. In particular, [6–8] have investigated the generalized XY model defined by $H = \sum_{\langle i,j \rangle} (1 - \Delta) \cos(\theta_i - \theta_j) + \Delta \cos(2\theta_i - 2\theta_j)$ where $\langle i, j \rangle$ denotes the sum over nearest neighbors. Reference [6] has shown how this model can be mapped onto a gas that is very similar to ours. The only difference resides in the pair interaction $V_\alpha(r)$, which is logarithmic in $r$ and anisotropic (because of $v$) in our model, but linear in $r$ and isotropic in the generalized XY model. Whether our gas maps onto a generalized XY model with local interactions remains an open question.

---

[2]Note that the gas temperature $T_{\text{gas}}$ is not uniquely defined since it always appears multiplied by some potential. Setting $T_{\text{gas}}$ amounts to fixing the energy scale, e.g. $\varepsilon_0$ for the Coulomb interaction.

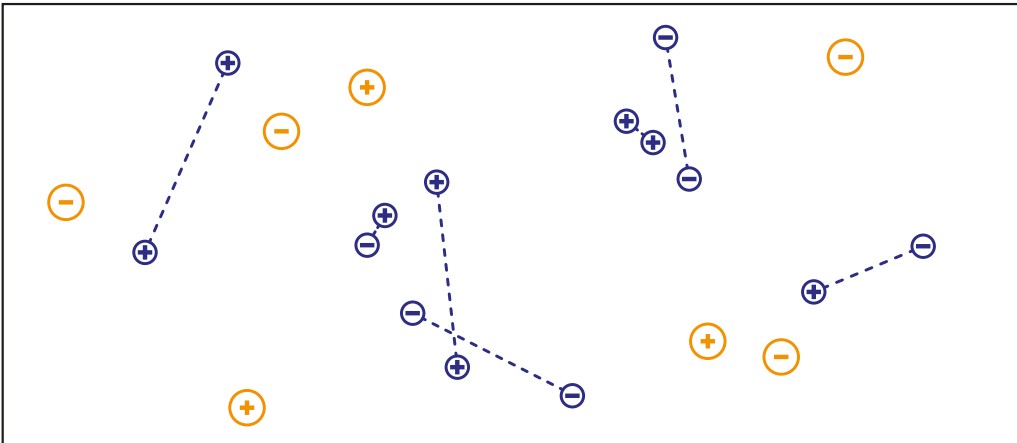

Figure 2: Snapshot of the generalized Coulomb gas. The large yellow circles are the $g^{\pm}$ particles with charge $\pm 1$. The small dark blue circles have charge $\pm 1/2$ and form the $\alpha$ pairs. The dark blue dashed lines represent the extra interaction $V_{\alpha}$ that comes from the bath kernel.

# 5 BKT-like phase transition

This section derives the phase diagram of the generalized Coulomb gas. The first subsection shows that the phase transition can be understood as the unbinding of charges, and the second derives perturbative renormalization group equations based on a low density expansion of the gas. We find that the gas undergoes a BKT-like phase transition at $K_c = \max(1 - s/2, 1/2)$ between a Luttinger liquid ($K > K_c$) and a dissipative phase ($K < K_c$).

## 5.1 Charge unbinding

It is known that the standard Coulomb gas undergoes a transition from a phase where all charges are in neutral bound groups, to a phase where charges are free to move [47]. The interpretation of the transition as the unbinding of charges (or, equivalently, of topological defects in the XY model) is a hallmark of the BKT phase transition. In this subsection we show that our generalized Coulomb gas possesses such a charge unbinding phase transition.

Before locating the phase transition, one must identify the possible phases. If all particles are tightly bound in neutral groups, coarse-graining the gas will fuse and annihilate the particles of a same group, leading to a gas with no particles. Therefore, this phase should be characterized, after coarse-graining, by vanishing fugacities of all particles ($z_g, z_{\alpha^+}, z_{\alpha^0}, z_{\alpha^-} \to 0$) and a renormalized permittivity ($\varepsilon_0 \to \varepsilon_R \varepsilon_0$). In the original field theory picture, this phase corresponds to a Luttinger liquid with renormalized velocity and Luttinger parameter ($u, K \to u_R, K_R$). On the contrary, as soon as a particle species becomes free, coarse-graining the field theory will not suppress all charges and the Luttinger liquid is destroyed. Free particles therefore heavily screen the Coulomb interaction which acquires an exponential decay (see the next subsection for more details).

Let us now describe the different bindings that occur in the gas. In the following we only present the bindings with the lowest particle number, so as to keep the argument concise, but a complete discussion of $n$-body bindings can be found in appendix C. There are three different types of binding to consider: the $g^+$-$g^-$ two-particle binding, the $\alpha^0$ two-particle binding, and the $\alpha^+$-$\alpha^-$ four-particle binding. In the following we review each scenario based on a heuristic free energy argument similar to that of [47] for the standard Coulomb gas.

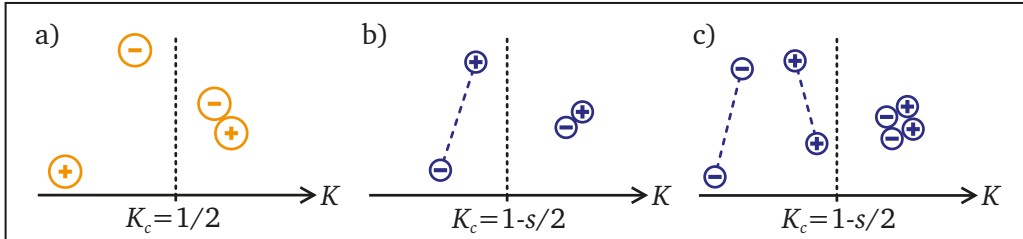

Figure 3: The lowest particle number bindings that occur in the generalized Coulomb gas. a) $g^+$-$g^-$ two-particle binding at $K_c = 1/2$. b) $\alpha^0$ two-particle binding at $K_c = 1-s/2$. c) $\alpha^+$-$\alpha^-$ four-particle binding at $K_c = 1-s/2$.

**$g^+$-$g^-$ two-particle binding.** Suppose there exists a $g^+$ and a $g^-$ particle in a region of typical length $R$ in all directions. It's binding energy $U$ comes solely from the Coulomb interaction in Eq. (22). Defining $U = -\int_{a<|r|<R} d^2r\, V_c(r) / \int_{a<|r|<R} d^2r$, one finds that $U \simeq T_{\text{gas}} 8K \log R/a$ at leading order. Moreover, since both particles occupy a region of area $\sim R^2$ and have a size $\sigma = a^2$, the entropy $S$ of the two particles is $S = 4\log(R/a)$. This leads to the free energy

$$F = U - T_{\text{gas}} S = T_{\text{gas}}(8K - 4)\log(R/a). \tag{24}$$

Minimizing the free energy $F$ shows the existence of a critical point at $K_c = 1/2$. If $K < K_c$, $F$ is minimized by a large $R$ and the particles are free, while if $K > K_c$, $F$ favors a small $R$ which means the particle pair binds (see inset a) in fig. 3). This transition is the BKT transition of the standard Coulomb gas.

**$\alpha^0$ two-particle binding.** Consider now an $\alpha^0$ pair with charge $(+1/2, -1/2)$. The binding energy $U$ comes from the Coulomb interaction $V_C$ and the bath-induced interaction $V_\alpha$, so $U \simeq T_{\text{gas}}(2K + 2 + s)\log(R/a)$ at leading order. Since the entropy is still $S = 4\log(R/a)$, the free energy of the pair is

$$F = T_{\text{gas}}(2K + s - 2)\log(R/a), \tag{25}$$

which gives a transition at $K_c = 1-s/2$. As $K$ is increased and crosses $K_c$, the particles go from being free to binding (see inset b) in fig. 3).

**$\alpha^+$-$\alpha^-$ four-particle binding.** Because an $\alpha^+$ particle pair (resp. $\alpha^-$ particle pair) has charge $(+1/2, +1/2)$ (resp. $(-1/2, -1/2)$) it cannot form on its own a neutral bound state. This is why we now consider an $\alpha^+$ pair and a $\alpha^-$ pair. The binding energy of the 4 particles is $U \simeq T_{\text{gas}}(4K + 4 + 2s)\log(R/a)$ while its entropy is $S = 8\log(R/a)$. The free energy is thus

$$F = T_{\text{gas}}(4K + 2s - 4)\log(R/a), \tag{26}$$

which gives a transition at $K_c = 1-s/2$. Below $K_c$, particles are free while, above $K_c$, they bind in groups of 4 particles (see inset c) in fig. 3).

The phase diagram inferred from these binding processes is depicted in fig. 4. Although this simple argument on the free energy misses the renormalization of the Luttinger liquid parameter $K$ and velocity $u$, it still captures the correct transition point for $\alpha, g \to 0$ as seen from the renormalization group equations derived in the next subsection.

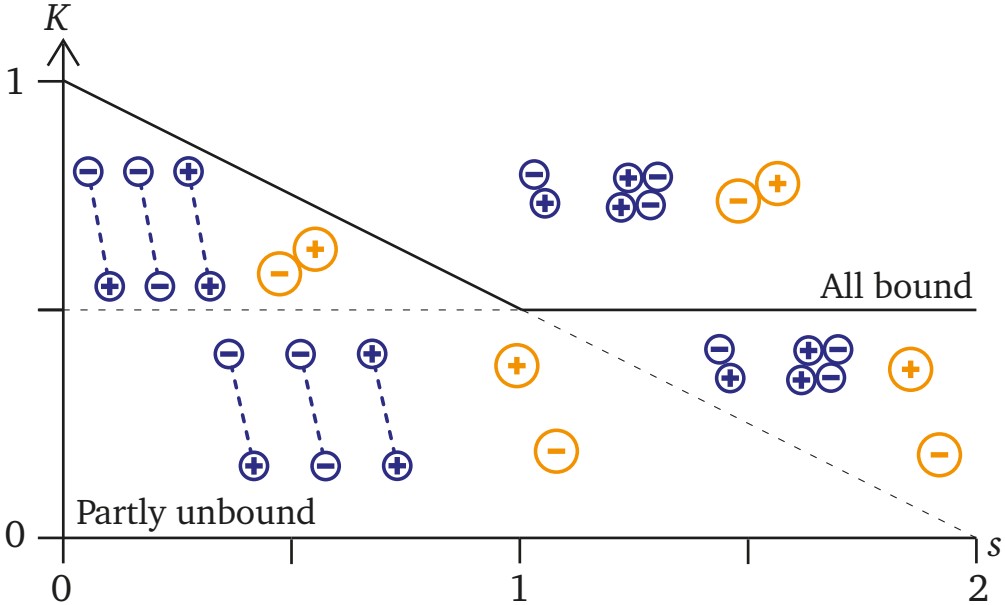

Figure 4: Phase diagram inferred from the binding of particles. The binding groups depicted here are those with the smallest particle number. The large yellow circles are the $g^{\pm}$ particles while the pairs of small dark blue circles form the $\alpha$ pairs. From the RG analysis, it appears there are only two phases: a Luttinger liquid where particles are all bound, and a dissipative phase which encompasses all three partly unbound regions.

## 5.2 RG equations in the Coulomb gas picture

We now derive the renormalization group (RG) equations of the generalized Coulomb gas. These equations will be perturbative in $\alpha$ and $g$ which translates to a low-fugacity, and thus low-density, expansion of the gas. Physically, in the absence of particles, the Coulomb interaction is characterized by the vacuum permittivity $\varepsilon_0$. When adding back particles, we expect the Coulomb interaction to be screened, leading to a renormalized permittivity $\varepsilon_R \varepsilon_0 > \varepsilon_0$. Since $K \propto \varepsilon_0^{-1}$, finite values of $\alpha$ and $g$ should yield a renormalized $K_R < K$. More formally, one can prove that (see appendix D)

$$K_R(\hat{q}) = K \left( 1 + 8\pi K \sum_i \hat{q}_i^2 \int d^2 r \, r_i^2 \langle n_c(r) n_c(0) \rangle \right), \tag{27}$$

where $\hat{q} = q/|q|$ with the frequency vector $q = (k, \omega_n/u)$. The determination of $K_R$ thus reduces to that of $\langle n_c(r) n_c(0) \rangle$. As anticipated, this density-density correlation function can be computed in a low density expansion valid for small $\alpha$ and $g$. At low density, the leading contribution to $\langle n_c(r) n_c(0) \rangle$ is given by two-particle configurations with particles at $0$ and $r$ weighted by the associated Boltzmann factor, i.e.

- an $\alpha^0$ pairing with the $+1/2$ charge at the origin and the $-1/2$ charge at position $r$ (or vice versa). Keeping track of the charges $(+1/2, -1/2)$, of the two possible configurations, and of the particle size $\sigma$,

$$\langle n_c(r) n_c(0) \rangle = \frac{2(-1/2)1/2}{\sigma^2} \frac{z_{\alpha^0}}{Z_{\text{gas}}} e^{\left( -\beta_{gas} \left( -\frac{1}{2} \right) \frac{1}{2} V_c(r) - \beta_{gas} V_\alpha(r) \right)} = -\frac{\alpha}{8\pi^2 a^4} \mathcal{D}(r) \left| \frac{a}{r} \right|^{2K}, \tag{28}$$

where we have used $Z_{\text{gas}} = 1 + \mathcal{O}(\alpha, g)$.

- a $g^+$ particle at the origin and a $g^-$ particle at position $r$ (or vice versa), which leads to

$$\langle n_c(r) n_c(0) \rangle = \frac{2(-1)1}{\sigma^2} \frac{z_g^2}{Z_{\text{gas}}} e^{-\beta_{gas}(-1)1V_c(r)} = -\frac{g^2}{8\pi^4 a^4} \left| \frac{a}{r} \right|^{8K} . \tag{29}$$

Combining these results with Eq. (27) yields (at order $\mathcal{O}(\alpha, g)$)

$$K_R^{-1}(\hat{q}) = K^{-1} + \frac{\alpha}{\pi} \sum_i \hat{q}_i^2 \int \frac{\mathrm{d}^2 r}{a^2} \frac{r_i^2}{a^2} \mathcal{D}(r) \left| \frac{a}{r} \right|^{2K} + \frac{g^2}{2\pi^3} \int \frac{\mathrm{d}^2 r}{a^2} \frac{r^2}{a^2} \left| \frac{a}{r} \right|^{8K} . \tag{30}$$

Noticing that $K_R(\hat{q} = (1,0)) = \frac{u}{u_R} K_R$ and $K_R(\hat{q} = (0,1)) = \frac{u_R}{u} K_R$, the RG equations for $u, K, \alpha, g$ can be derived from Eq. (30) by varying the short distance cutoff $a \to a e^{\mathrm{d}l}$ (see appendix E). These equations read

$$\frac{\mathrm{d}K^{-1}}{\mathrm{d}l} = \frac{g^2}{\pi^2} + \frac{\alpha}{2\pi} F_K \left( \frac{u}{v} \right), \tag{31}$$

$$\frac{\mathrm{d}u}{\mathrm{d}l} = \frac{\alpha u K}{2\pi} F_u \left( \frac{u}{v} \right), \tag{32}$$

$$\frac{\mathrm{d}\alpha}{\mathrm{d}l} = (2 - s - 2K)\alpha, \tag{33}$$

$$\frac{\mathrm{d}g}{\mathrm{d}l} = (2 - 4K)g, \tag{34}$$

where

$$F_K(x) = \int_0^{2\pi} \mathrm{d}\theta \left( \sin^2 \theta + x^2 \cos^2 \theta \right)^{-1-s/2}, \tag{35}$$

$$F_u(x) = \int_0^{2\pi} \mathrm{d}\theta \, \cos(2\theta) \left( \sin^2 \theta + x^2 \cos^2 \theta \right)^{-1-s/2} . \tag{36}$$

Looking at these equations, it is clear that a phase transition occurs at $K_c = \max(1 - s/2, 1/2)$, as expected from the binding argument given in the previous subsection. For $K > K_c$, both $\alpha$ and $g$ are irrelevant and the system flows to a Luttinger liquid, i.e. an empty gas, under the RG. For $K < K_c$, $g$ (if $s < 1$) or $\alpha$ (if $s > 1$) becomes relevant and the Luttinger liquid is unstable.

From Eqs. (31-34), it seems that a second phase transition occurs at $K_c' = \min(1/2, 1 - s/2)$ since both $\alpha$ and $g$ are relevant when $K < K_c'$. It turns out there is actually only one phase for $K < K_c$ as pointed out by [33] where a similar problem was studied. In hand waving terms, as soon as, for example, $\alpha$ becomes relevant, $\alpha$ pairs start to proliferate in the gas. However, at long distances, an $\alpha^\pm$ pair cannot be distinguished from a $g^\pm$ particle so $g$ particles start proliferating too. Formally, this should appear as a $\mathcal{O}(\alpha)$ additional term in Eq. (34). Similarly, when $g$ particles start proliferating, coarse-graining can combine them with an $\alpha$ pair, e.g. $g^+ + \alpha^- \to \alpha^0$. This should appear as a $\mathcal{O}(g\alpha)$ term in Eq. (33).

Having understood that there is a unique dissipative phase for $K < K_c$, one may ask what its nature is. Based on the well-known phase diagram of the standard Coulomb gas, we expect the dissipative phase to have a strongly screened Coulomb interaction, with an exponential fallout at large distance. The characteristic length $\xi$ of this decay can be inferred from the RG equation (33). Let us consider the RG flow up to a point where the dissipative effects become dominant, i.e. $\xi(l^\star) \sim 1$. Owing to the scaling dimension of $\xi$, its microscopic value $\xi(l = 0)$ is related to $\xi(l^\star)$ as

$$\xi(l = 0) = \xi(l^\star) e^{l^\star} \sim e^{l^\star} . \tag{37}$$

To evaluate $l^\star$ one must distinguish between three cases: superohmic baths ($s > 1$), ohmic baths ($s = 1$), and subohmic baths ($s < 1$).

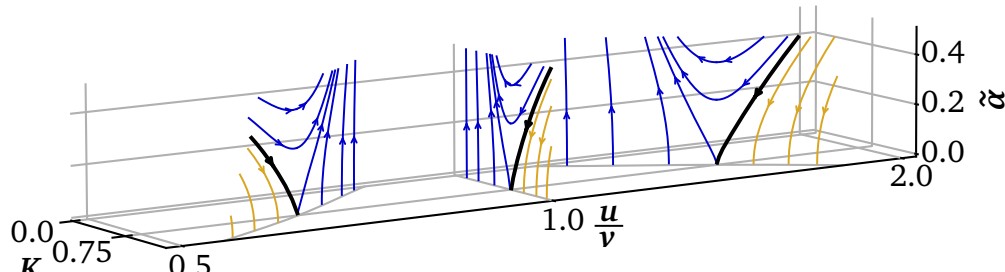

Figure 5: Subohmic ($s = 0.5$) RG flow of the couplings $(K, u/v, \tilde{\alpha})$ from a numerical integration of Eqs. (41). The Luttinger liquid phase is depicted in gold while the dissipative phase is blue. The transition line in the $\alpha = 0$ plane is at $K_c = 1 - s/2 = 0.75$ for all $u/v$. The $u/v$ axis is in log scale.

**Superohmic bath.** For $s > 1$, the transition is located at $K_c = 1/2$. Near this transition point, $\alpha$ has scaling dimension $1 - s < 0$ and is strongly irrelevant. The coupling $\alpha$ can thus be dropped and the remaining RG equations become

$$\frac{\mathrm{d}K^{-1}}{\mathrm{d}l} = \frac{g^2}{\pi^2}, \qquad \frac{\mathrm{d}u}{\mathrm{d}l} = 0, \qquad \frac{\mathrm{d}g}{\mathrm{d}l} = (2 - 4K)g, \tag{38}$$

which are exactly those of the BKT transition in the sine-Gordon model [43, 47]. This RG flow is isotropic since $u$ is not renormalized, and the correlation function $\langle e^{i(\phi(r) - \phi(0))} \rangle = \left| \frac{a}{r} \right|^\eta$ exhibits the anomalous dimension $\eta = K_c/2 = 1/4$. It is possible to access the dissipation length in the limit of $g \to 0$ since $g(l^\star) \simeq g(l = 0)e^{(2-4K)l^\star}$ and $l^\star$ is defined as $g(l^\star) \sim 1$. This leads to

$$\xi(l = 0) \sim g(l = 0)^{-\frac{1}{2-4K}}, \tag{39}$$

which, as expected, diverges when $K$ tends to $K_c = 1/2$.

**Ohmic bath.** For $s = 1$, $\alpha$ and $g$ both become relevant at $K_c = 1/2$ and must be treated simultaneously. Writing the RG equations in terms of $\tilde{\alpha} = \sqrt{\alpha}$ yields

$$\frac{\mathrm{d}K^{-1}}{\mathrm{d}l} = \frac{g^2}{\pi^2} + \frac{\tilde{\alpha}^2}{2\pi} F_K\left(\frac{u}{v}\right), \quad \frac{\mathrm{d}u}{\mathrm{d}l} = \frac{\tilde{\alpha}^2 uK}{2\pi} F_u\left(\frac{u}{v}\right), \quad \frac{\mathrm{d}\tilde{\alpha}}{\mathrm{d}l} = \left(\frac{1}{2} - K\right)\tilde{\alpha}, \quad \frac{\mathrm{d}g}{\mathrm{d}l} = (2 - 4K)g, \tag{40}$$

which shows that $\tilde{\alpha}$ and $g$ play quite similar roles. The only significant difference is that $\tilde{\alpha}$ can break the isotropy of the system by renormalizing $u$. Writing the RG equations in terms of $\tilde{\alpha}$ is very natural when thinking in the Coulomb gas picture. Since $\alpha$ was the fugacity of an $\alpha$ particle pair (up to a numerical prefactor), $\tilde{\alpha}$ appears as the fugacity of a single particle of the pair and is the right quantity to compare with $g$. The transition at $s = 1$ is thus a BKT-like transition with $\eta = 1/4$ and is driven by two couplings.

**Subohmic bath.** For $s < 1$, the coupling $g$ is irrelevant near $K_c = 1 - s/2$ since its scaling dimension is $2s - 2 < 0$. Neglecting this coupling, the RG equations in terms of $\tilde{\alpha} = \sqrt{\alpha}$ reduce to

$$\frac{\mathrm{d}K^{-1}}{\mathrm{d}l} = \frac{\tilde{\alpha}^2}{2\pi} F_K\left(\frac{u}{v}\right), \qquad \frac{\mathrm{d}u}{\mathrm{d}l} = \frac{\tilde{\alpha}^2 uK}{2\pi} F_u\left(\frac{u}{v}\right), \qquad \frac{\mathrm{d}\tilde{\alpha}}{\mathrm{d}l} = \left(1 - \frac{s}{2} - K\right)\tilde{\alpha}, \tag{41}$$

which is similar to a BKT transition but with an anomalous dimension $\eta = K_c/2 = (2 - s)/4$ and a renormalized velocity $u$. Hence the name BKT-like phase transition. The associated RG flow is depicted in figure 5 for $s = 0.5$. Notice how each of the three panels looks like the typical BKT flow. The panel at $u/v = 1$ describes an isotropic system which, as expected,

remains isotropic after coarse-graining (i.e. $u(l)/v = 1$). The two side panels at $u/v \neq 1$ show that, in both phases, the RG flow always tends to bring $u/v$ closer to 1, reducing the system's anisotropy. As for the subohmic case, it is possible to compute the dissipation length in the limit of $\tilde{\alpha} \to 0$ since $\tilde{\alpha}(l^\star) \simeq \tilde{\alpha}(l=0)e^{(1-s/2-K)l^\star}$ and $l^\star$ is such that $\tilde{\alpha}(l^\star) \sim 1$. This yields

$$\xi(l=0) \sim \tilde{\alpha}(l=0)^{-\frac{1}{1-s/2-K}}, \tag{42}$$

which diverges when $K$ goes to $K_c = 1 - s/2$.

## 6 Discussion and conclusion

In this work, we have discussed the sine-Gordon to Coulomb gas mapping in the case of an XXZ spin chain at zero magnetization and subject to spatially correlated dissipation. This dissipation was introduced by adding baths of phonons, *à la* Caldeira and Leggett, which were linearly coupled to one another to allow for spatial correlations. This dissipative setup has, to the best of the author's knowledge, not been studied before. At low energy, it turns out this microscopic system can be mapped through bosonization onto a dissipative sine-Gordon action with long-range interactions. The resulting field theory has a very natural interpretation in terms of a generalized Coulomb gas. On top of the usual particles of the Coulomb gas with charge $\pm 1$, our model presents particles with charge $\pm 1/2$ and connected in pairs by a charge-independent logarithmic interaction. In this Coulomb gas picture, we identify a BKT-like phase transition from the binding-unbinding transition, a signature of the BKT transition, and from its RG equations near the critical point. The location of the BKT transition depends on the type of bath studied through the bath exponent $s$. For superohmic baths ($s > 1$), the transition is the usual BKT transition caused by the binding of $\pm 1$ charges. However, for subohmic baths ($s < 1$), the transition is shifted by dissipation and is due to the binding of the $\pm 1/2$ charges.

The procedure followed in this article can be straightforwardly extended to other one-dimensional quantum systems described by a sine-Gordon-like effective action. Studies of an XXZ spin chain, at zero [33] and finite magnetization [31, 32], subject to local baths (i.e. $v = 0$ in our model) can be readily understood in the gas picture. Adapting our analysis, one would find a dissipative kernel $\mathcal{K}(x, \tau) \propto \delta(x)/|\tau|^{1+s}$ (see Appendix A.2). This corresponds to bath-induced particle-pairs with coordinates $(x, u\tau)$ and $(x, u\tau')$ (notice the identical $x$ coordinates). While the presence of particle-pairs of fixed orientation might seem to be an artifact specific to this model, it is actually a feature shared by all field theories endowed with the *statistical tilt symmetry* (STS) or space-gauged shift symmetry, i.e. the invariance of the interacting part of the action $S - S_{\mathrm{LL}}$ under $\phi(x, \tau) \to \phi(x, \tau) + f(x)$. In the gas picture, the STS indeed requires that the total charge along any $x = \mathrm{cst}$ slice of the gas must be zero, thus strongly restricting the possible relative orientations of particles. This result is easily shown by considering the Boltzmann weight $\mathcal{W} \propto \langle \prod_j \exp[i4q_j \phi(r_j)] \rangle_{\mathrm{LL}}$ associated with a configuration of particles of charges $q_j$ at positions $r_j$. Using the STS, one gets that $\mathcal{W} \propto \exp[i4 \sum_j q_j f(x_j)] \langle \prod_j \exp[i4q_j \phi(r_j)] \rangle_{\mathrm{LL}}$ for all functions $f(x)$. This implies that allowed particle configurations must be neutral along any $x = \mathrm{cst}$ slice of the 2D space. The STS appears generically in the replicated action of one-dimensional systems with quenched disorder, as in the well-known Giamarchi-Schulz model [16, 19].

In summary, we have shown that the effect of dissipation on a spin chain can induce a BKT-like phase transition which can be easily understood in terms of a 2D generalized Coulomb gas.

## Acknowledgments

The author would like to thank Grégory Schehr for sparking the idea of this work and for useful discussions, Laura Foini and Alberto Rosso for useful discussions and careful proofreading of this manuscript, as well as Raphaël Dulac, Julien Moy and Nicolas Paris for being very useful rubber ducks, and Nicolas Dupuis for encouraging the writing of this paper.

## A  Bath kernel

### A.1  General derivation

Upon integrating out the bath degrees of freedom in Eqs. (10,11), we are left with the bath kernel

$$\mathcal{K}(x,\tau) = \sum_\gamma \frac{\lambda_\gamma^2}{m_\gamma} G_\gamma(x,\tau), \tag{A.1}$$

where $G_\gamma(x,\tau) = \frac{1}{2\pi v} K_0(\Omega_\gamma \sqrt{\tau^2 + \frac{x^2}{v^2}})$ is the Green's function of the imaginary time Klein–Gordon operator $-\partial_\tau^2 - v^2 \partial_x^2 + \Omega_\gamma^2$. To further simplify $\mathcal{K}(x,\tau)$, we use the following definition of the low-energy behavior of the spectral form factor

$$J(\Omega) = \frac{\pi}{2} \sum_\gamma \frac{\lambda_\gamma^2}{\Omega_\gamma m_\gamma} \delta(\Omega - \Omega_\gamma) = N\alpha \tau_c^{s-1} \Omega^s, \quad \text{for} \quad \Omega \in [0, 1/\tau_c], \tag{A.2}$$

where the numerical prefactor is $N^{-1} = \frac{u}{v\pi^2} \int_0^\infty dy\, y^{1+s} K_0(y)$. Note that $N$ is positive since $K_0(y) > 0$ for $y \in \mathbb{R}_+$, and well-defined for all $s \geq 0$ as $K_0(y) \overset{y \gg 1}{\sim} \sqrt{\frac{\pi}{2y}} e^{-y}$. Thus,

$$\begin{aligned}
\mathcal{K}(x,\tau) &= \frac{2}{\pi} \int d\Omega\, J(\Omega)\Omega \frac{1}{2\pi v} K_0\left(\Omega\sqrt{\tau^2 + x^2/v^2}\right) \\
&= \frac{N\alpha\tau_c^{s-1}}{\pi^2 v} \int_0^{1/\tau_c} d\Omega\, \Omega^{1+s} K_0\left(\Omega\sqrt{\tau^2 + x^2/v^2}\right) \\
&= \frac{\alpha\tau_c^{s-1}}{u\sqrt{\tau^2 + x^2/v^2}^{2+s}} \frac{\int_0^{\sqrt{\tau^2 + x^2/v^2}/\tau_c} dy\, y^{1+s} K_0(y)}{\int_0^\infty dy\, y^{1+s} K_0(y)} \\
&\simeq \frac{\alpha\tau_c^{s-1}}{u} \left(\tau^2 + x^2/v^2\right)^{-1-s/2}, \quad \text{for} \quad \sqrt{\tau^2 + x^2/v^2} \gg \tau_c.
\end{aligned} \tag{A.3}$$

Using this expression along with the constraint $a = u\tau_c$ yields the kernel that appears in Eq. (13).

### A.2  Uncorrelated and Markovian limits

We derive the kernels $\mathcal{K}(x,\tau)$ obtained in the limit of uncorrelated baths $v \to 0$ and in the Markovian limit $v \to \infty$. Let us first proceed with the $v \to 0$ limit. We start from the second line of Eq. (A.3),

$$\mathcal{K}(x,\tau) = \frac{N\alpha\tau_c^{s-1}}{\pi^2 v} \int_0^{1/\tau_c} d\Omega\, \Omega^{1+s} K_0\left(\Omega\sqrt{\tau^2 + x^2/v^2}\right), \tag{A.4}$$

and assume that $\tilde{\alpha} = \alpha v/u \propto \alpha N$ is kept constant when $v \to 0$ (which, from Eq. (A.2) just amounts to $J(\Omega)$ having a well-defined limit when $v \to 0$). If we let $v \to 0$ for $x \neq 0$, because

of the exponential decay of the modified Bessel function $K_0$ for large argument, we obtain $\lim_{v \to 0} \mathcal{K}(x \neq 0, \tau) = 0$, while it is obvious that for $x = 0$, $\lim_{v \to 0} \mathcal{K}(0, \tau) = +\infty$. To verify that $\mathcal{K}(x, \tau)$ behaves as a Dirac $\delta(x)$ distribution, we must calculate the weight

$$\int_{-\infty}^{+\infty} dx \mathcal{K}(x, \tau) = \frac{N \alpha \tau_c^{s-1}}{\pi^2} \int_0^{1/\tau_c} d\Omega \, \Omega^{1+s} \int_{-\infty}^{+\infty} \frac{dx}{v} K_0 \left( \Omega \sqrt{\tau^2 + x^2/v^2} \right)$$

$$= \frac{1}{|\tau|^{1+s}} \frac{N \alpha \tau_c^{s-1}}{\pi^2} \int_0^{|\tau|/\tau_c} dy \, y^{1+s} \int_{-\infty}^{+\infty} du \, K_0 \left( y \sqrt{1 + u^2} \right). \tag{A.5}$$

When $|\tau| \gg \tau_c$, we can extend the $y$ integration to $+\infty$ and use the definition of the normalization $N$ to find

$$\int_{-\infty}^{+\infty} dx \mathcal{K}(x, \tau) = \frac{\tilde{\alpha} \tau_c^{s-1}}{|\tau|^{1+s}} \frac{\int_0^{+\infty} dz \, z^{1+s} K_0(z) \int_{-\infty}^{+\infty} du (1 + u^2)^{-1-s/2}}{\int_0^{+\infty} dy \, y^{1+s} K_0(y)}$$

$$= \frac{\tilde{\alpha} \tau_c^{s-1}}{|\tau|^{1+s}} \frac{\sqrt{\pi} \Gamma(\frac{1+s}{2})}{\Gamma(1 + s/2)}. \tag{A.6}$$

Therefore, $\lim_{v \to 0} \mathcal{K}(x, \tau) \propto \delta(x) \tilde{\alpha} / |\tau|^{1+s}$ and one recovers the decay $1/|\tau|^{1+s}$ expected for independent baths.

The Markovian limit $v \to \infty$ is obtained in a similar fashion by computing the weight $\int d\tau K(x, \tau)$. One finds that

$$\int_{-\infty}^{+\infty} d\tau \mathcal{K}(x, \tau) = \frac{N \alpha \tau_c^{s-1} v^s}{\pi^2} \frac{1}{|x|^{1+s}} \int_0^{|x|/(v\tau_c)} dy \, y^{1+s} \int_{-\infty}^{+\infty} du K_0 \left( y \sqrt{1 + u^2} \right). \tag{A.7}$$

The interesting scaling limit is obtained by adopting the new UV cutoff $\tau_c' = \tau_c v/u$ and keeping constant $\tilde{\alpha} = \alpha (v/u)^2$ such that, for distances $|x| \gg u \tau_c'$,

$$\int_{-\infty}^{+\infty} d\tau \mathcal{K}(x, \tau) = \frac{\tilde{\alpha} \tau_c'^{s-1} u^s}{|x|^{1+s}} \frac{\sqrt{\pi} \Gamma(\frac{1+s}{2})}{\Gamma(1 + s/2)}, \tag{A.8}$$

which shows that the kernel reduces to $\lim_{v \to \infty} \mathcal{K}(x, \tau) \propto \delta(\tau) \tilde{\alpha} / |x|^{1+s}$.

# B  Properties of the Luttinger liquid action

In this section we state some properties of the Luttinger liquid action $S_{\text{LL}}$ introduced in Eq. (8) that we have used throughout the paper.

**Shift symmetry.**  This action enjoys the shift symmetry $S_{\text{LL}}[\phi(r) + \phi_0] = S_{\text{LL}}[\phi(r)]$ which implies that for any set of numbers $c_1, ..., c_n$

$$\forall \phi_0, \quad \left\langle e^{\sum_i c_i \phi(r_i)} \right\rangle_{\text{LL}} = e^{\phi_0 \sum_i c_i} \left\langle e^{\sum_i c_i \phi(r_i)} \right\rangle_{\text{LL}}$$

$$= 0, \quad \text{if } \sum_i c_i \neq 0. \tag{B.1}$$

This is the property which enforces charge neutrality in the generalized Coulomb gas.

**Propagator.**  The two-point function of $S_{\text{LL}}$ is given by

$$\langle \phi(r) \phi(r') \rangle_{\text{LL}} = -\frac{K}{2} \log \left| \frac{r - r'}{a} \right|, \tag{B.2}$$

where $a$ is a short-distance cut-off.

## C  General $n$-body binding

Let us consider a neutral group of particles with particle numbers $n_{g^+}, n_{g^-}, n_{\alpha^+}, n_{\alpha^0}, n_{\alpha^-}$. We wish to compute the threshold $K_c$ for the binding of such a group. If one supposes that all particles are roughly at a distance $R$ from one another, then the total Coulomb interaction energy is given at leading order by

$$
U_c = -T_{\text{gas}} 8K \log(R/a)\Bigg[ \frac{n_g^+(n_g^+ - 1)}{2} + \frac{n_g^-(n_g^- - 1)}{2} - \frac{n_{\alpha^0}}{4} + \frac{n_{\alpha^+}(2n_{\alpha^+} - 1)}{4} + \frac{n_{\alpha^-}(2n_{\alpha^-} - 1)}{4}
$$
$$
- n_{g^-}n_{g^+} + n_{g^+}n_{\alpha^+} + n_{g^-}n_{\alpha^-} - n_{g^-}n_{\alpha^+} - n_{g^+}n_{\alpha^-} - n_{\alpha^+}n_{\alpha^-} \Bigg], \quad \text{(C.1)}
$$

the total energy of the dissipative interaction is

$$
U_\alpha = T_{\text{gas}}(2+s)\log(R/a)(n_{\alpha^+} + n_{\alpha^0} + n_{\alpha^-}), \quad \text{(C.2)}
$$

and the total entropy is

$$
S = \log(R/a)(2n_{g^+} + 2n_{g^-} + 4n_{\alpha^+} + 4n_{\alpha^0} + 4n_{\alpha^-}). \quad \text{(C.3)}
$$

Putting everything together, the total free energy $F = U_C + U_\alpha - T_{\text{gas}}S$ amounts to

$$
\frac{F}{T_{\text{gas}}\log(R/a)} = -8K\Bigg[ \frac{n_{g^+}(n_{g^+} - 1)}{2} + \frac{n_{g^-}(n_{g^-} - 1)}{2} - \frac{n_{\alpha^0}}{4} + \frac{n_{\alpha^+}(2n_{\alpha^+} - 1)}{4} + \frac{n_{\alpha^-}(2n_{\alpha^-} - 1)}{4}
$$
$$
- n_{g^-}n_{g^+} + n_{g^+}n_{\alpha^+} + n_{g^-}n_{\alpha^-} - n_{g^-}n_{\alpha^+} - n_{g^+}n_{\alpha^-} - n_{\alpha^+}n_{\alpha^-} \Bigg]
$$
$$
+ (2+s)(n_{\alpha^+} + n_{\alpha^0} + n_{\alpha^-}) - (2n_{g^+} + 2n_{g^-} + 4n_{\alpha^+} + 4n_{\alpha^0} + 4n_{\alpha^-})
$$
$$
= (4K - 2)(n_{g^+} + n_{g^-}) + (2K + s - 2)(n_{\alpha^+} + n_{\alpha^0} + n_{\alpha^-})
$$
$$
- 4K(n_{g^+} - n_{g^-} + n_{\alpha^+} - n_{\alpha^-})^2. \quad \text{(C.4)}
$$

Recall that the binding group must be neutral, i.e. $n_{g^+} - n_{g^-} + n_{\alpha^+} - n_{\alpha^-} = 0$. The free energy thus reduces to

$$
F = nT_{\text{gas}}\log(R/a)\big[(4K - 2)(x_{g^+} + x_{g^-}) + (2K + s - 2)(x_{\alpha^+} + x_{\alpha^0} + x_{\alpha^-})\big], \quad \text{(C.5)}
$$

where we have introduced the total number of particles $n = \sum_i n_i$ and the particle fractions $x_i = \frac{n_i}{n}$. Since we are working we the free energy $F$, we must minimize it over the parameters $R, \{x_i\}$ while keeping the total number $n$ of particles fixed. This leads to distinguishing between four cases.

- If $K > 1/2$ and $K < 1 - s/2$, all contributions to $F$ are minimized by taking $R \to 0$. This is the case of *full binding*, i.e. any neutral particle group can form a bound state. Notice that, technically speaking, one should talk of groups of particles which do not repel from one another, rather than binding groups. Indeed, for instance, we call here a set of $g^+, g^+, g^-, g^-$ particles a binding group although it actually forms two $g^+ - g^-$ groups which do not interact as they are neutral.

- If $s < 1$ and $1 - s/2 > K > 1/2$, then $F$ is minimized by setting either i) $x_{g^+} = x_{g^-} = 0$ and $R \to 0$, or ii) $x_{\alpha^+} = x_{\alpha^0} = x_{\alpha^-} = 0$ and $R \to \infty$. This means that $\alpha$ pairs bind while $g$ particles do not, signaling *partial binding*. Notice that charge neutrality imposes an equal number of $\alpha^+$ and $\alpha^-$ pairs in bound groups. The possible bound states with smallest particle number are an $\alpha^0$ two-particle group and an $\alpha^+ - \alpha^-$ four-particle group as stated in the main text.

- If $s > 1$ and $1/2 > K > 1 - s/2$, the minimization of free energy requires setting either i) $x_{\alpha^+} = x_{\alpha^0} = x_{\alpha^-} = 0$ and $R \to 0$, or ii) $x_{g^+} = x_{g^-} = 0$ and $R \to \infty$. This corresponds to *partial binding* where only $g$ particles bind. Again, the possible bound state with smallest particle number is the $g^+ - g^-$ two-particle group which is presented in the main text.

- If $K < 1/2$ and $K < 1 - s/2$, all contributions to $F$ are minimized by taking $R \to \infty$. This is the case of *no binding*, i.e. all particles are free.

## D  Electromagnetic screening

This appendix gives a proof of Eq. (27) following the lines of [45]. Let us start by performing a Hubbard-Stratonovitch transformation on Eq. (21) to rewrite the partition function $Z_{\text{gas}}$ as

$$
\begin{aligned}
Z_{\text{gas}} = \sum_{n=0}^{\infty} \sum_{\{n_i\}}^{(n)} & \frac{z_g{}^{n_{g^+} + n_{g^-}} z_{\alpha^+}{}^{n_{\alpha^+}} z_{\alpha^0}{}^{n_{\alpha^0}} z_{\alpha^-}{}^{n_{\alpha^-}}}{n_{g^+}! n_{g^-}! n_{\alpha^+}! n_{\alpha^0}! n_{\alpha^-}!} \\
& \times \prod_j \int \frac{d^2 r_j}{\sigma} \exp\left[ -\frac{1}{2} \beta_{\text{gas}} \int d^2 r \, d^2 r' \, n_\alpha(r, r') V_\alpha(r - r') \right] \\
& \times \int \mathcal{D}[\varphi] \exp\left[ -\frac{1}{32\pi K} \int d^2 r (\nabla \varphi)^2 + i \int d^2 r \, n_c(r) \varphi(r) \right].
\end{aligned}
\tag{D.1}
$$

The additional field $\varphi$ is the scalar potential of electromagnetism if one defines the vacuum permittivity $\varepsilon_0 = (16\pi K)^{-1}$. Introducing this extra field is a useful trick as, for $n_c(r) = 0$ its propagator is $\langle \varphi(q) \varphi(q') \rangle = \frac{K}{q^2} 64\pi^3 \delta^{(2)}(q + q')$, which for $n_c(r) \neq 0$ then gives a direct definition of $K_R(q)$ through

$$
\langle \varphi(q) \varphi(q') \rangle = \frac{K_R(q)}{q^2} 64\pi^3 \delta^{(2)}(q + q'),
\tag{D.2}
$$

where $q = (k, \omega_n/u)$ and $q' = (k', \omega'_n/u)$. Finding $K_R(q)$ therefore amounts to computing the scalar potential propagator. To do so, one introduces an external current $J(r)$ that couples to the scalar potential as

$$
Z_{\text{gas}}[J] = \sum_{n=0}^{\infty} [\cdots] \int \mathcal{D}[\varphi] \exp\left[ -\frac{1}{32\pi K} \int d^2 r (\nabla \varphi)^2 + \int d^2 r [i n_c(r) + J(r)] \varphi(r) \right],
\tag{D.3}
$$

and defines

$$
\langle \varphi(q) \varphi(q') \rangle = 16\pi^4 \left. \frac{\delta^2 \log Z[J]}{\delta J(-q) \delta J(-q')} \right|_{J=0}.
\tag{D.4}
$$

Integrating out the field $\varphi$ from $Z_{\text{gas}}[J]$ and partially going to Fourier space then yields

$$
\begin{aligned}
Z_{\text{gas}}[J] = \sum_{n=0}^{\infty} \sum_{\{n_i\}}^{(n)} & \frac{z_g{}^{n_{g^+} + n_{g^-}} z_{\alpha^+}{}^{n_{\alpha^+}} z_{\alpha^0}{}^{n_{\alpha^0}} z_{\alpha^-}{}^{n_{\alpha^-}}}{n_{g^+}! n_{g^-}! n_{\alpha^+}! n_{\alpha^0}! n_{\alpha^-}!} \\
& \times \prod_j \int \frac{d^2 r_j}{\sigma} \exp\left[ -\frac{1}{2} \beta_{\text{gas}} \int_{|r-r'|>a} d^2 r \, d^2 r' \left( n_c(r) V_c(r - r') n_c(r') + n_\alpha(r, r') V_\alpha(r - r') \right) \right] \\
& \times \exp\left[ \int \frac{d^2 q}{4\pi^2} \frac{8\pi K}{q^2} J(q) J(-q) + \int \frac{d^2 q}{4\pi^2} i \frac{16\pi K}{q^2} n_c(q) J(-q) \right].
\end{aligned}
\tag{D.5}
$$

Since we want to take the second derivative of $Z_{\text{gas}}[J]$ with respect to the source $J$, we only need to compute $Z_{\text{gas}}[J]$ up to $\mathcal{O}(J^2)$. Calling $\langle\cdots\rangle$ the thermodynamic average with $J = 0$, and performing a cumulant expansion up to $\mathcal{O}(J^2)$ yields

$$
\begin{aligned}
\frac{Z_{\text{gas}}[J]}{Z_{\text{gas}}[0]} &= \left\langle \exp\left[ \int \frac{d^2q}{4\pi^2}\frac{8\pi K}{q^2}J(q)J(-q) + \int \frac{d^2q}{4\pi^2}i\frac{16\pi K}{q^2}n_c(q)J(-q) \right]\right\rangle \\
&= \exp\left[ \int \frac{d^2q}{4\pi^2}\frac{d^2q'}{4\pi^2}\frac{8\pi K}{q^2}J(q)J(q')\left(4\pi^2\delta^{(2)}(q+q') - \frac{16\pi K}{q'^2}\langle n_c(-q)n_c(-q')\rangle\right) \right],
\end{aligned}
\tag{D.6}
$$

which leads to

$$
\langle\varphi(q)\varphi(q')\rangle = \frac{16\pi K}{q^2}\left(4\pi^2\delta^{(2)}(q+q') - \frac{16\pi K}{q^2}\langle n_c(-q)n_c(-q')\rangle\right).
\tag{D.7}
$$

The total system being invariant under translation, the density-density average is given by $\langle n_c(q)n_c(q')\rangle = 4\pi^2\delta^{(2)}(q+q')\chi_{\text{nn}}(q)$, where $\chi_{\text{nn}}(q) = \int d^2r\, e^{-iq\cdot r}\langle n_c(r)n_c(0)\rangle$. Hence the exact result

$$
K_R(q) = K\left(1 - \frac{16\pi K}{q^2}\chi_{\text{nn}}(q)\right),
\tag{D.8}
$$

which shows that the charges screen the bare Coulomb interaction. Since we are only interested in the renormalization of the velocity $u$ and the Luttinger parameter $K$ which describe the Gaussian part of the original field theory, it suffices to compute $\chi_{\text{nn}}(q)$ to order $\mathcal{O}(q^2)$. This amounts to dropping any irrelevant operator scaling as $q^3, q^4, \ldots$ in the action. Starting from the definition of $\chi_{\text{nn}}(q)$ and expanding it yields

$$
\begin{aligned}
\chi_{\text{nn}}(q) &= \int d^2r(1 - iq\cdot r - \frac{(q\cdot r)^2}{2} + \ldots)\langle n_c(r)n_c(0)\rangle \\
&= -\frac{1}{2}\sum_i q_i^2 \int d^2r\, r_i^2\langle n_c(r)n_c(0)\rangle,
\end{aligned}
\tag{D.9}
$$

where we have used the fact that the gaz is neutral, i.e. $\int d^2r\, n_c(r) = 0$, and invariant under the parity transformation $r_i \to -r_i$. Putting everything together finally yields

$$
K_R(\hat{q}) = K\left(1 + 8\pi K\sum_i \hat{q}_i^2 \int d^2r\, r_i^2\langle n_c(r)n_c(0)\rangle\right), \qquad \hat{q} = q/|q|,
\tag{D.10}
$$

which is Eq. (27) in the text.

# E RG equations

In this appendix, we derive the RG equations given in Eq. (31-34). We begin by recalling Eq. (30),

$$
K_R^{-1}(\hat{q}) = K^{-1} + \frac{\alpha}{\pi}\sum_i \hat{q}_i^2 \int_{|r|>a} \frac{d^2r}{a^2}\frac{r_i^2}{a^2}\mathcal{D}_{\tau_c}(r)\left|\frac{a}{r}\right|^{2K} + \frac{g^2}{2\pi^3}\int_{|r|>a} \frac{d^2r}{a^2}\frac{r^2}{a^2}\left|\frac{a}{r}\right|^{8K},
\tag{E.1}
$$

where the short-distance cutoff in the integrals and in $\mathcal{D}_{\tau_c}$ has been made explicit. Notice how the cutoffs in the integrals are $\sqrt{x^2 + (u\tau)^2} > a$ and not $\sqrt{x^2 + (v\tau)^2} > a$. The first cutoff is that of the Coulomb interaction, while the second comes from the bath-induced interaction. As any choice of UV cutoff should qualitatively yield the same low-energy physics, we decide

to stick to the Coulomb one. One then uses $\mathcal{D}_{\tau_c}(x,\tau) = \left[(\tau/\tau_c)^2 + (u/v)^2(x/a)^2\right]^{-1-s/2}$ and splits the integrals at $a' = ae^{dl}$ (and $\tau_c' = \tau_c e^{dl}$ since $a = u\tau_c$) as follows

$$
\begin{aligned}
K_R^{-1}(\hat{q}) &= K^{-1} + \frac{g^2}{2\pi^3} \int\limits_{|r|>a'} \frac{d^2r}{a^2} \frac{r^2}{a^2} \left|\frac{a}{r}\right|^{8K} + \frac{g^2}{2\pi^3} \int\limits_{a'>|r|>a} \frac{d^2r}{a^2} \frac{r^2}{a^2} \left|\frac{a}{r}\right|^{8K} \\
&+ \frac{\alpha}{\pi} \sum_i \hat{q}_i^2 \int\limits_{|r|>a'} \frac{d^2r}{a^2} \frac{r_i^2}{a^2} \mathcal{D}_{\tau_c}(r) \left|\frac{a}{r}\right|^{2K} + \frac{\alpha}{\pi} \sum_i \hat{q}_i^2 \int\limits_{a'>|r|>a} \frac{d^2r}{a^2} \frac{r_i^2}{a^2} \mathcal{D}_{\tau_c}(r) \left|\frac{a}{r}\right|^{2K} \\
&= K^{-1} + \frac{g^2}{2\pi^3} \int\limits_{|r|>a'} \frac{d^2r}{a'^2} \frac{r^2}{a'^2} \left|\frac{a'}{r}\right|^{8K} e^{dl(4-8K)} + \frac{g^2}{2\pi^3} \int\limits_{e^{dl}>|\xi|>1} d^2\xi \\
&+ \frac{\alpha}{\pi} \sum_i \hat{q}_i^2 \int\limits_{|r|>a'} \frac{d^2r}{a'^2} \frac{r_i^2}{a'^2} \mathcal{D}_{\tau_c'}(r) \left|\frac{a'}{r}\right|^{2K} e^{dl(2-s-2K)} + \frac{\alpha}{\pi} \sum_i \hat{q}_i^2 \int\limits_{e^{dl}>|\xi|>1} \frac{d^2\xi\, \xi_i^2}{\left[\xi_2^2 + \left(\frac{u}{v}\right)^2 \xi_1^2\right]^{1+s/2}} \\
&= K^{-1}(\hat{q},dl) + \frac{\alpha(dl)}{\pi} \sum_i \hat{q}_i^2 \int\limits_{|r|>a'} \frac{d^2r}{a'^2} \frac{r_i^2}{a'^2} \mathcal{D}_{\tau_c'}(r) \left|\frac{a'}{r}\right|^{2K} + \frac{g(dl)^2}{2\pi^3} \int\limits_{|r|>a'} \frac{d^2r}{a'^2} \frac{r^2}{a'^2} \left|\frac{a'}{r}\right|^{8K},
\end{aligned}
\tag{E.2}
$$

where we have introduced the variable $\xi = (\xi_1, \xi_2) = (x/a, \tau/\tau_c)$ and defined

$$
K^{-1}(\hat{q}, dl) = K^{-1} + \frac{g^2}{2\pi^3} \int\limits_{e^{dl}>|\xi|>1} d^2\xi + \frac{\alpha}{\pi} \sum_i \hat{q}_i^2 \int\limits_{e^{dl}>|\xi|>1} d^2\xi\, \xi_i^2 \left[\xi_2^2 + \left(\frac{u}{v}\right)^2 \xi_1^2\right]^{-1-s/2}, \tag{E.3}
$$

as well as $\alpha(dl) = \alpha e^{dl(2-s-2K)}$ and $g(dl) = g e^{dl(2-4K)}$. Notice that Eq. (E.2) is the same as Eq. (30) we started with but with renormalized quantities. Taking $dl \to 0$ in the previous renormalized quantities directly leads to the RG equations for $\alpha$ and $g$

$$
\frac{d\alpha}{dl} = (2 - s - 2K)\alpha, \tag{E.4}
$$

$$
\frac{dg}{dl} = (2 - 4K)g. \tag{E.5}
$$

The RG equations for $K$ and $u$ are extracted by plugging $K(\hat{q} = (1,0), dl) = \frac{u}{u(dl)} K(dl)$ and $K(\hat{q} = (0,1), dl) = \frac{u(dl)}{u} K(dl)$ in Eq. (E.3), which leads to (with $\xi = (\cos\theta, \sin\theta)$)

$$
\frac{1}{u} \frac{d}{dl} \frac{u}{K} = \frac{g^2}{\pi^2} + \frac{\alpha}{\pi} \int_0^{2\pi} d\theta\, \cos^2\theta \left[\sin^2\theta + \left(\frac{u}{v}\right)^2 \cos^2\theta\right]^{-1-s/2}, \tag{E.6}
$$

$$
u \frac{d}{dl} \frac{1}{uK} = \frac{g^2}{\pi^2} + \frac{\alpha}{\pi} \int_0^{2\pi} d\theta\, \sin^2\theta \left[\sin^2\theta + \left(\frac{u}{v}\right)^2 \cos^2\theta\right]^{-1-s/2}. \tag{E.7}
$$

Separating $u$ and $K$ finally yields the two last RG equations

$$
\frac{dK^{-1}}{dl} = \frac{g^2}{\pi^2} + \frac{\alpha}{2\pi} \int_0^{2\pi} d\theta \left[\sin^2\theta + \left(\frac{u}{v}\right)^2 \cos^2\theta\right]^{-1-s/2}, \tag{E.8}
$$

$$
\frac{du}{dl} = \frac{\alpha uK}{2\pi} \int_0^{2\pi} d\theta\, \cos(2\theta) \left[\sin^2\theta + \left(\frac{u}{v}\right)^2 \cos^2\theta\right]^{-1-s/2}. \tag{E.9}
$$

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
