# Peer review of "Mapping a dissipative quantum spin chain onto a generalized Coulomb gas"

_SciPost Physics, doi:SciPost Phys. 17, 130 (2024)_

## Round 1 · Referee Report · Anonymous (Referee 1) · 2024-5-3

Strengths

  • derivation of a mapping of the Caldeira-Leggett action of a quantum sine-Gordon model coupled to a harmonic oscillator bath to a classical generalized Coulomb gas.
  • renormalization group study of the Berezinskii-Kosterlitz-Thouless transition of the generalized Coulomb gas

Report

The article reports on a mapping of a XXZ spin-1/2 chain interacting with a phonon bath on a generalized classical Coulomb gas. The mapping allows to identify a Berezinskii-Kosterlitz-Thouless transition between a Tomonaga-Luttinger liquid phase and a dissipative phase in the case of superohmic dissipation using renormalization group equations. With ohmic dissipation, the BKT transition occurs at the same point as the transition to the Ising antiferromagnet phase, and in the subhomic case, it is preempted by the formation of the Ising antiferromagnet. The article is very clearly written, with all the necessary details of derivations described in Appendices A-E.
The abstract and main results section describe the main findings which are also exposed in the first paragraph of the conclusions. The relevant previous work is covered in Refs. [21-34] and Ref. [38].
The article establishes an interesting link between quantum phase transitions in the presence of dissipation induced by a harmonic bath and classical statistical mechanics of generalized Coulomb gases. Phase transitions in dissipative quantum mechanics have undergone a recent regain of attention from theoretician, and a connection with a well studied subject such as Coulomb gas is likely to yield quick progress. An aspect which is not touched in the paper, but considered in Refs. [30,31] for a different model is the characterization of the phases in which dissipation dominates. The present article is likely to stimulate such studies. I thus recommend publication in SciPost.

Recommendation

Publish (easily meets expectations and criteria for this Journal; among top 50%)

---

## Round 1 · Referee Report · Anonymous (Referee 2) · 2024-5-31

Strengths

1- exact mapping between a dissipative spin chain and a generalized Coulomb gas 2- RG equations are solved for sub- and superohmic baths 3- well written and well explained 4- an intuitive mechanism for the dissipation-driven quantum phase transition is developed

Report

Quantum many-body systems coupled to an environment have gained increasing interest in the last years and the coupling to a bath can drive interesting quantum phase transitions. In this work, the author considers a one-dimensional XXZ chain where each local spin operator $S^{z}_i$ is coupled to an independent dissipative bosonic bath. The paper describes a mapping from a bosonized action for the dissipative chain to a generalized Coulomb gas. In particular, the Coulomb-gas approach provides an intuitive picture for the dissipation-induced unbinding transition, which includes a new set of particles originating from a bath-induced long-range interaction. One of the main results is the difference between sub- and superohmic baths which become relevant at different Luttinger parameters.

This work provides a link between a quantum many-body problem that is of current interest and a stat-mech problem that has an intuitive understanding. The paper includes sufficient details to follow the derivation, but also puts a lot of emphasis in describing the physical picture in simple words. I enjoyed reading the paper and learned a lot from the discussion. All in all, I recommend publication in SciPost Physics.

Requested changes

1- I am wondering about the decay of the dissipation kernel $D(x,\tau)$ in Eq. (14) and its dependence on the bath exponent $s$. For $\nu=0$ it becomes $D(x,\tau)\propto \delta(x)/(\tau/\tau_c)^{2+s}$. In particular, for an ohmic bath with $s=1$ this leads to a retarded interaction that decays as $1/\tau^3$. I always thought that an ohmic bath corresponds to a temporal decay $\propto 1/\tau^2$, as it is also the case in Ref. [33] (a closely related paper by the author). It would be good to clarify this difference to the author's previous work (or correct it if it is a mistake).

2- On page 8 [below Eq. (20)], the author writes $D(r-r)$. Isn't it just $D(r)$?

3- First line on page 11 [above Eq. (24)]: The formatting of the in-line equation $U=\dots$ is not clear because of the minus sign in front of $V_c(r)$. Can one just put it in front of the integral?

4- On page 12 [the line below Eq. (27)]: I assume it is $K_R$ instead of $K_r$, right?

Recommendation

Publish (easily meets expectations and criteria for this Journal; among top 50%)

---

## Round 2 · Referee Report · Anonymous (Referee 1) · 2024-8-26

Strengths

same as in previous report

Report

In my previous report, I had already recommended publication of the manuscript in SciPost. A few misprints numbered 2-4 were pointed out by the other referee and have been corrected in the revised version.
The second referee also raised the issue of the limit $\nu \to 0$ (point 1) and the authors have responded that the Green's function of the bath operator $(-\nu^2 \partial_x^2 -\partial_\tau^2+\Omega^2)^{-1}$ does not deform confinuously into the one on the local bath operator $(-\partial_\tau^{2} +\Omega^2)^{-1}$. They have also inserted a footnote in the manuscript making that argument.

I am not sure the argument of the authors is the correct one.
If we insert Eqs. (5) into Eq. (12) we find
\begin{eqnarray}
\label{eq:douze}
\mathcal{K}(x,\tau)=\frac{1}{\pi^2 \nu} \int_0^{\tau_c^{-1}} d\Omega \alpha \tau_c^{s-1} \Omega^{s+1} K_0\left(\Omega \sqrt{\tau^2 + (x/\nu)^2} \right)
\end{eqnarray}
If we let $\nu \to 0$ for $x\ne 0$, because of the exponential decay of the modified Bessel function for large argument, we obtain
\begin{equation}
\lim_{\nu \to 0} \mathcal{K}(x\ne 0,\tau) =0,
\end{equation}
while it is obvious that for $x=0$,
\begin{equation}
\lim_{\nu \to 0} \mathcal{K}(0,\tau) =+\infty.
\end{equation}
To verify that $\mathcal{K}(x,\tau)$ behaves as a Dirac delta distribution, we only need to calculate the weight
\begin{eqnarray}
\label{eq:weight}
\int dx \mathcal{K}(x,\tau) &=& \frac{1}{\pi^2} \int_0^{\tau_c^{-1}} d\Omega \alpha \tau_c^{s-1} \Omega^{s+1} \int_{-\infty}^{+\infty} K_0\left(\Omega \sqrt{\tau^2 + (x/\nu)^2} \right) \frac{dx}{\nu}, \\
&=& \frac{\alpha \tau_c^{s-1}}{\pi^2 \tau^{s+1}} \int_0^{\tau/\tau_c} dw w^{s+1} \int_{-\infty}^{+\infty} du K_0 (w \sqrt{1+u^2}),
\end{eqnarray}
and note that it is independent of $\nu$. Moreover, when $\tau \gg \tau_c$, we can extend the $w$ integration to $+\infty$ to find
\begin{eqnarray}
\label{eq:asymp-w}
\int dx \mathcal{K}(x,\tau\gg \tau_c) &\sim& \frac{\alpha \tau_c^{s-1}}{\pi^2 \tau^{s+1}} \int_{-\infty}^{+\infty} \frac{du}{(1+u^2)^{1+s/2}} \int_0^{+\infty} dv v^{s+1} K_0(v) \\
&\sim& \frac{\alpha \tau_c^{s-1}}{\pi^2 \tau^{s+1}} \\
&\sim& \frac{\alpha \tau_c^{s-1}\Gamma(s+1)}{\pi \tau^{s+1}},
\end{eqnarray}
and recover for $s=1$ the decay $1/\tau^2$ expected in an ohmic bath.
So even though the bath operators do not deform continuously into each other,
it is possible to recover the correct limit for $\nu \to 0$ directly from Eq. (12).
In other words, the comment 1 of the other referee was incorrect, and the authors
have been too cautious in their response and footnote.

I suggest the authors replace their footnote with a derivation of the delta distribution limit of the kernel from Eq. (12) and that final version of the manuscript gets published in SciPost.

Attachment

Recommendation

Publish (easily meets expectations and criteria for this Journal; among top 50%)

---

## Round 2 · Referee Report · Anonymous (Referee 2) · 2024-9-5

Report

I thank the author for clarifying on page 7 that the $\nu\to0$ limit cannot be taken trivially from the derived equations. The author also discusses the implications of the $\nu\to0$ limit in Section 6. Because this discussion is rather technical, the meaning is not fully clear to me. As I understand this part, the Coulomb gas picture applies here as well, with the only difference that the interaction in Eq. (23) becomes local in space (and remains nonlocal in time). Is that correct?

In the introduction, the author reviews how the Coulomb-gas picture had been applied to generalized XY models, but I do not find any discussion of applications to dissipative systems. In the past, dissipation effects had also been studied in the context of Josephson junctions. For example, the Coulomb gas picture is mentioned for a single dissipative junction [Schmid, Phys. Rev. Lett. 51, 1506 (1983)] or for arrays of dissipative junctions [Bobbert, Fazio, Schön, Zimanyi, Phys. Rev. B 41, 4009 (1990)], but probably also in other works. It would be fair to review also relevant work including dissipation and to check if there are any similarities to the present work.

Recommendation

Ask for minor revision

---

## Round 2 · Author Response

Dear Editor,

I would like to thank both the referees for their valuable comments and appreciation of the results shown in the article. Following their suggestions and comments, I have edited certain parts of the manuscript. Below, I present a point-by-point response to all the queries of the referees.
Yours sincerely,

Oscar Bouverot-Dupuis

---

## Round 2 · List of Changes

"1- I am wondering about the decay of the dissipation kernel D(x,τ) in Eq. (14) and its dependence on the bath exponent s. For ν=0 it becomes D(x,τ)∝δ(x)/(τ/τc)2+s. In particular, for an ohmic bath with s=1 this leads to a retarded interaction that decays as 1/τ3. I always thought that an ohmic bath corresponds to a temporal decay ∝1/τ2, as it is also the case in Ref. [33] (a closely related paper by the author). It would be good to clarify this difference to the author's previous work (or correct it if it is a mistake)."

Answer : Thank you for pointing out this issue. The bath Kernel D(x,τ) is, roughly speaking, the Green's function of the bath operator G−1. For local baths, G−1 is the one dimensional operator G−11=−∂2τ+Ω2, while for non-local bath it is the two dimensional operator G−12=−v2∂2x−∂2τ+Ω2. It turns out that, as v is taken to 0, the Green's function G2 does not go to G1. This means that to study the v=0 case, one cannot directly set v=0 in the bath kernel D(x,τ) written in the article, but one has rather to rederive its expression from the microscopic model with v=0. With this in mind, local ohmic baths do indeed correspond to a temporal decay ∝1/τ2. Page 7 has been corrected accordingly and a footnote has also been added.

"2- On page 8 [below Eq. (20)], the author writes D(r−r). Isn't it just D(r)?"

3- First line on page 11 [above Eq. (24)]: The formatting of the in-line equation U=… is not clear because of the minus sign in front of Vc(r). Can one just put it in front of the integral?

4- On page 12 [the line below Eq. (27)]: I assume it is KR instead of Kr, right?"

Thanks for spotting these typos, they have been corrected.

---

## Round 3 · Author Response

Dear Editor,

I would like to thank both referees for submitting a second round of valuable comments on my article. Following their suggestions and comments, I have edited certain parts of the manuscript. Below, I present a point-by-point response to all the queries of the referees.
Yours sincerely,

Oscar Bouverot-Dupuis

---

## Round 3 · List of Changes

Response to referee #1: 1) The second referee also raised the issue of the limit ν→0 (point 1) and the authors have responded that the Green's function of the bath operator (−ν2∂2x−∂2τ+Ω2)−1 does not deform confinuously into the one on the local bath operator (−∂2τ+Ω2)−1. [...] I suggest the authors replace their footnote with a derivation of the delta distribution limit of the kernel from Eq. (12) and that final version of the manuscript gets published in SciPost. Answer : I would like to sincerely thank the referee for making what I believe is a very valuable addition to the manuscript. I have put the derivation of the ν→0 limit in appendix A.2 and referred to it in the last paragraph of section 3.

Response to referee #2: "1) The author also discusses the implications of the ν→0 limit in Section 6. Because this discussion is rather technical, the meaning is not fully clear to me. As I understand this part, the Coulomb gas picture applies here as well, with the only difference that the interaction in Eq. (23) becomes local in space (and remains nonlocal in time). Is that correct?" Answer : Yes, that is fully correct. In order to make this clearer, I have rewritten the beginning of the second paragraph of Section 6.

2) In the introduction, the author reviews how the Coulomb-gas picture had been applied to generalized XY models, but I do not find any discussion of applications to dissipative systems. In the past, dissipation effects had also been studied in the context of Josephson junctions. For example, the Coulomb gas picture is mentioned for a single dissipative junction [Schmid, Phys. Rev. Lett. 51, 1506 (1983)] or for arrays of dissipative junctions [Bobbert, Fazio, Schön, Zimanyi, Phys. Rev. B 41, 4009 (1990)], but probably also in other works. It would be fair to review also relevant work including dissipation and to check if there are any similarities to the present work." Answer: I thank the referee for pointing these articles out. I have also found some relevant references in Monte-Carlo studies of dissipative quantum systems. I have included them in the second to last paragraph of section 1, starting from "It is worth mentioning that...". I also discuss the similarities and discrepancies with my work.

---

## Editorial Decision

published